# Prevalence of Migraine and Neuropathic Pain in Rheumatic Diseases

**DOI:** 10.3390/jcm9061890

**Published:** 2020-06-17

**Authors:** Sylvain Mathieu, Marion Couderc, Bruno Pereira, Jean-Jacques Dubost, Sandrine Malochet-Guinamand, Anne Tournadre, Martin Soubrier, Xavier Moisset

**Affiliations:** 1Service de Rhumatologie, Université Clermont-Auvergne, CHU Gabriel Montpied, 63000 Clermont-Ferrand, France; mcouderc@chu-clermontferrand.fr (M.C.); jjdubost@chu-clermontferrand.fr (J.-J.D.); smalochet@chu-clermontferrand.fr (S.M.-G.); atournadre@chu-clermontferrand.fr (A.T.); msoubrier@chu-clermontferrand.fr (M.S.); 2Rheumatology Department, Gabriel Montpied Teaching Hospital, 58 Rue Montalembert, 63003 Clermont-Ferrand, France; 3Unité de biostatistiques (DRCI), CHU Clermont-Ferrand, 63000 Clermont-Ferrand, France; bpereira@chu-clermontferrand.fr; 4Neurology Department, Université Clermont Auvergne, CHU de Clermont-Ferrand, Inserm, Neuro-Dol, F-63000 Clermont Ferrand, France; xmoisset@chu-clermontferrand.fr

**Keywords:** rheumatic diseases, migraine, neuropathic pain

## Abstract

To investigate the physiopathology of pain in chronic inflammatory rheumatic diseases (CIRDs), we assessed the prevalence of migraine and neuropathic pain in 499 patients with CIRDs. We studied 238 patients with rheumatoid arthritis, 188 with spondyloarthritis (SpA), 72 with psoriatic arthritis (PsA), and 1 unclassified. Migraine was diagnosed according to IHS migraine diagnostic criteria. Neuropathic pain was diagnosed when patients scored at least 3 on the DN4 questionnaire. Participants completed a validated self-assessment questionnaire. Migraine prevalence was 34% (165/484), and it was highest in PsA. Risk factors for migraine were a high level of anxiety, female sex, young age, and TNF-alpha inhibitor treatment (OR = 1.90 (1.13–3.25)). Besides, high disease activity was a risk factor in SpA. Blood CRP level was not significantly associated with migraine. Of 493 patients with CIRDs, 21.5% had chronic pain with neuropathic characteristics. Compared to the French general population, these patients had significantly higher prevalences of migraine (two-fold) and neuropathic pain (three-fold). This study showed that migraine and neuropathic pain frequently occurred in patients with rheumatic diseases. Therefore, upon reporting residual pain, these patients should be checked for the presence of migraine or neuropathic pain, despite adequate clinical control of rheumatic disease.

## 1. Introduction

Chronic inflammatory rheumatoid diseases (CIRDs) are a class of rheumatic diseases that includes rheumatoid arthritis (RA), spondyloarthritis (SpA), and psoriatic arthritis (PsA). The common feature of these diseases is that they cause inflammatory pain, which leads to functional disability. The current therapeutic objective is to achieve clinical remission, that is, the absence of pain, joint swelling, spinal stiffness, and biological inflammatory syndrome [1,2]. However, although patients are in clinical remission, some have reported persistent pain. The most likely hypothesis is that the nociceptive pathways are sensitized and endogenous controls of nociception are altered in CIRDs, a chronic algogenic disease. This condition might be described as the presence of a fibromyalgia state associated with CIRDs. Clauw et al. reported that 10–30% of patients with RA, osteoarthritis, or lupus met the criteria for fibromyalgia [3]. The prevalence of fibromyalgia was estimated to be 6 to 17% in RA and 4 to 15% in spondyloarthritis [4,5,6]. In a previous study, we found that 24% of 143 patients with CIRDs met the criteria for fibromyalgia [7]. However, a recent study showed that the frequency of neuropathic pain, defined as pain arising as a direct consequence of a lesion or disease affecting the somatosensory system, was relatively high among patients with RA, PsA, or SpA, without necessarily meeting the criteria for fibromyalgia [8]. Therefore, persistent pain in patients with CIRDs might not be linked to the presence of fibromyalgia; instead, it might involve other parameters. The physiopathology of pain in rheumatic diseases remains unknown; indeed, few studies have evaluated the frequency of neuropathic pain in CIRDs. In one study, Koop et al. found that, among 159 patients with RA, 27 (17%) had likely neuropathic pain and 34 (21.4%) had possible neuropathic pain [9]. Similarly, although many studies have described a link between chronic inflammatory activity and the risk of migraine in multiple sclerosis and inflammatory bowel disease [10,11], very few studies have described the occurrence of migraine in CIRDs. In the general French population, the prevalence of migraine is 21.3% and the prevalence of chronic neuropathic pain is 6.9% [12,13]. Although there is no pathophysiological relationship between migraine and neuropathic pain, it has been shown in a previous study focusing on pain in patients with multiple sclerosis that these two pain entities were associated [10]. As there was a rationale for an increased prevalence of both migraine and neuropathic pain types in CIRDs, we decided to study both in the same study.

This study aimed to assess the prevalence of migraine and neuropathic pain in patients with CIRDs that were routinely followed in a Rheumatology department. We then compared these prevalences with those found in the general French population.

## 2. Methods

### 2.1. Ethical Considerations

This study was approved by the local Ethics Committee (*Comité de Protection des Personnes Sud-Ouest et Outre-Mer II)* (n°ID-RCB: 2017-A01655-48). All included patients provided written consent to participate in the study, in accordance with the Declaration of Helsinki.

### 2.2. Patients

All patients who visited the Rheumatology Department of Clermont-Ferrand University Hospital for RA, SpA, or PsA were invited to participate in the study. The following criteria were used to classify patients: The 2010 American College of Rheumatology/European League Against Rheumatism criteria for RA and the Classification Criteria for Psoriatic Arthritis for PsA. For SpA, we used the New York criteria for ankylosing spondylitis or the axial Assessment of SpondyloArthritis International Society (ASAS) criteria for non-radiographic spondyloarthropathy or the peripheral ASAS criteria for peripheral spondyloarthropathy.

We collected clinical and demographic data, including age, gender, tobacco use, type of rheumatic disease, and year of diagnosis. Disease activity was assessed with the Disease Activity Score in 28 joints, based on the erythrocyte sedimentation rate (DAS28 ESR), for RA or PsA, and with the Bath Ankylosing Spondylitis Disease Activity Index (BASDAI) for SpA and PsA. Functional disability was evaluated with the health assessment questionnaire (HAQ). Anxiety and depression were evaluated with the hospital anxiety and depression (HAD) questionnaire [14]. All participants also completed the pain catastrophizing scale [15]. Systemic inflammation was evaluated based on the blood CRP level. We also recorded comorbidities that could cause chronic pain, such as osteoarthritis, obesity, Gougerot–Sjogren syndrome, fibromyalgia, and current treatments. Each patient completed a self-assessment questionnaire validated in previous studies to report migraine and/or neuropathic pain [10,16].

### 2.3. Survey

To ensure a maximal response rate, the questionnaire was deliberately simple; it was designed to be completed in less than 15 min for most of the patients. Validated questionnaires were used to limit the risk of bias. All participants first responded to the question: “Are you experiencing or have you experienced pain or headache during the last 3 months?”. Participants who responded “No” skipped the remainder of the pain questionnaire, and they completed the sections on anxiety, depression, and disease activity. Participants who responded “yes” completed specific surveys that covered headaches and other types of pain. After completion, the questionnaires were checked for accuracy by one of the investigators.

Headaches. The first part of the questionnaire served to identify migraine; it included the diagnostic criteria for a strict migraine established in the third edition of the International Classification of Headache Disorders [17]. This questionnaire was similar to a self-administered questionnaire used previously in French surveys to diagnose strict and probable migraines [12]. Migraines were diagnosed according to the following International Headache Society (IHS) migraine diagnostic criteria: Typical headache that lasted 4–72 h without treatment, at least two of four typical headache characteristics (unilateral, pulsatile, pain intensity ≥4/10 on the visual analogue scale (VAS) for pain, increase in pain with physical activity), at least one of two types of nonpain-associated symptoms (nausea and/or vomiting, photophobia, and phonophobia). Participants were diagnosed with a strict migraine when their attacks met all the IHS diagnostic criteria for migraines. Participants were diagnosed with a probable migraine when their attacks met all but one of the four diagnostic criteria for migraines without an aura. With this method, a previous study estimated that migraine prevalence was 21.3% in the general population in France [12]. This diagnostic questionnaire was followed by a six-item, short-form survey for measuring headache impact (HIT-6) [18]. Chronic migraine is characterized by the presence of ≥15 days of headache per month for at least three months, with headache having the same clinical features of migraine without aura for at least eight of those 15 days in the absence of medication overuse [17]. In the present study, those with migraine who experienced headache ≥15 days per month were considered to have chronic migraine, as proposed in previous studies [10,16].

Other pain types. A grading system is recommended for neuropathic pain’s diagnosis [19]. Although this is the gold standard, such a grading system can be difficult to use for nonspecialists and using validated screening questionnaires in large epidemiological studies is commonly accepted. In the present study, the subjects answered the two questions (including seven items) from the DN4 interview questionnaire regarding the characteristics of their pain [13]. A score of 1 was given to each positive item and a score of 0 to each negative item. The total score was calculated as the sum of the seven items. Respondents with a total score ≥3 were considered to have neuropathic pain characteristics [13]. Initially, the DN4 interview questionnaire was validated as a clinician-administered questionnaire. A complementary validation was made, and the results of the self-reported and clinician-administered questions for each of the seven items showed excellent consistency (κ coefficients with a 95% confidence interval (CI) of 0.82–0.95, *p* < 0.001). The self-administered DN4 questionnaire had a sensitivity of 81.6% and a specificity of 85.7% for an optimal cut-off score of 3 out of 7, being similar to those observed in the initial study using a clinician-administered version of the questionnaire [20]. With this method, a previous study estimated that the prevalence of chronic neuropathic pain was 6.9% in the general population in France [13]. Participants also completed the brief pain inventory (BPI) [21], which measured both pain intensity and whether pain interfered with everyday life.

### 2.4. Statistical Analysis

The primary objective of this study was to assess the prevalence of migraine and neuropathic pain in a sample of patients with inflammatory rheumatic diseases. Continuous data are presented as the mean ± standard deviation (SD) or the median and interquartile range (IQR), based on the statistical distribution. The assumption of normality was checked with normal probability plots and the Shapiro–Wilk test. Prevalence and categorical data are presented as the percentage and 95% confidence interval (95% CI).

Continuous parameters were compared between independent groups (RA/SpA/PsA, with/without migraine, with/without neuropathic pain) with the ANOVA or student *t*-test. Alternatively, when the assumptions of parametric tests were not met, we used the Kruskal–Wallis test or Mann–Whitney test. The homoscedasticity hypothesis was verified with the Bartlett test. Categorical data were compared between groups with the chi-squared test or Fisher’s exact test. Binary variables of particular interest, like the presence/absence of migraine or pain, were evaluated with generalized linear models (i.e., logistic regression), which included these variables as response variables and estimated the odds ratios (OR) and corresponding 95% CI. Furthermore, we performed multivariable analyses to make simultaneous adjustments for variables that showed univariate significance or clinical relevance.

We compared the prevalences of migraine and neuropathic pain in our sample to those of the general population, based on Lanteri and Bouhassira studies. Differences were expressed as ORs, based on the Mantel–Haenszel method [12,13].

All statistical analyses were performed with R and STATA software. All tests were two-sided, and the type I error was set at 5%. Following Rothman, we chose to report all the individual *p*-values and confidence intervals, without doing any mathematical correction for distinct tests comparing two modalities (Rothman, K.J. Epidemiology 1990).

## 3. Results

We included 499 patients with CIRDs (238 with RA, 188 with SpA, 72 with PsA, and one patient not classified). Patient characteristics are summarized in Table 1 and Figure 1.

Nearly two-thirds of patients 313/488 (64.1%) reported that they experienced pain. Active disease was particularly prevalent in the PsA group (47%). In contrast, in the RA group, the mean DAS28 ESR (2.34 ± 1.23) corresponded to disease remission. The entire population showed low systemic inflammation (median CRP = 3.0 mg/L, IQR: 1.3–6.0) and low rates of fibromyalgia (6%) and depression (8%). Nearly half of the patients were treated with methotrexate or Tumor Necrosis Factor (TNF)-alpha inhibitors, particularly for RA and SpA, respectively.

The prevalence of migraine was 34% (165/484) and the prevalence of strict migraine was 13.2% (64/484). These prevalences were higher in the PsA group. Parameters associated with a higher prevalence of migraine are shown in Table 2, Table 3 and Table 4.

Female sex and a higher level of anxiety were always associated with migraine, regardless of the type of rheumatic disease. Younger patients had a higher risk of migraine in the entire population and in the RA group. High disease activity (BASDAI, OR = 1.03, 95% CI: 1.01–1.05) was associated with the prevalence of migraine only in the SpA group. Systemic inflammation was never significantly associated with migraine. TNF-alpha blockade treatment was associated with a significantly higher risk of migraine (OR = 1.90, 95% CI: 1.13–3.25). Other treatments were not associated with the occurrence of migraine. Among the 165 patients with migraine, 124 (75.2%) had HIT-6 scores above 55, which indicated that headache had an important impact on these patients. In addition, the prevalence of chronic migraine was very high (12%). We found no difference in the headache burden among the three CIRDs groups (Table 5).

Among 493 patients with inflammatory rheumatic diseases, 21.5% had chronic neuropathic pain characteristics. The intensity of pain was moderate (5.0 ± 1.5). Prevalence was significantly higher in the SpA group (26.7%) than in the PsA (19.4%) and RA (17.6%) groups (*p* = 0.02). Active disease (OR = 1.00, 95% CI: 1.00–1.03), and salazopyrine treatment (OR = 1.90. 95% CI: 1.05–12.7) were associated with a higher risk of neuropathic pain (Table 6). Moreover, neuropathic pain was associated with migraine (*p* < 0.001) (Table 2). For active disease, we observed a significant *neuropathic pain x migraine* (strict and probable) interaction (*p* = 0.02). This interaction was highlighted by the higher DAS28 ESR scores observed in patients with neuropathic pain and migraine (2.98 ± 1.36) compared to patients without migraine or neuropathic pain (2.18 ± 0.93), patients with neuropathic pain but no migraine (2.26 ± 0.91), and patients with migraine but no neuropathic pain (2.21 ± 0.92). However, the *neuropathic pain x migraine* interaction was not a significant factor in the HAQ (*p* = 0.33), HAD anxiety (*p* = 0.74), PCS (*p* = 0.99), or HAD depression (*p* = 0.65) tests.

Compared to the general population, our cohort showed significantly higher rates of migraine (OR = 1.91, 95% CI: 1.57–2.32) and neuropathic pain (OR = 3.71, 95% CI: 2.97–4.62), based on the Mantel–Haenszel method. Similarly, we observed significant differences in strict and probable migraine rates between the general population and our cohort, when we performed separate analyses, according to gender (OR = 1.74, 95% CI: 1.38–2.19 for females and OR = 1.75, 95% CI: 1.20–2.54 for males) and age (OR = 2.05, 95% CI:1.18–3.57 for ages <35 years; OR = 2.52, 95% CI: 1.87–3.40, for ages 35–54 years; and OR = 2.38, 95% CI: 1.74–3.25, for ages >54 years). However, we found no significant difference in the effect of gender (i.e., female vs. male) on the prevalence of migraines between our patients (OR = 2.34, 95% CI: 1.54–3.64) and the general population (OR = 2.36, 95% CI: 2.14–2.60; *p* = 0.96). The effects of age (i.e., ages 35–54 years vs. >54 years and ages <35 years vs. >54 years) on the prevalence of migraines were also similar between our patients (respectively, OR = 2.92, 95% CI: 1.93–4.42 and OR = 2.67, 95% CI: 1.43–4.98) and the general population (respectively, OR = 2.75, 95% CI: 2.43-3.13; *p* = 0.44; and OR = 3.09, 95% CI: 2.71–3.53; *p* = 0.19). The higher frequency of neuropathic pain in our sample compared to the general population was more dramatic among younger patients (OR = 7.00, 95% CI: 3.70; 13.24, for ages <35 years; OR = 6.23, 95% CI: 4.47; 8.68, for ages 35–54 years; and OR = 1.72, 95% CI: 1.21; −2.45, for ages >54 years), but was not different between the sexes (OR = 3.52, 95% CI: 2.70; 4.57 for females; and OR = 3.43, 95%CI: 2.28; 5.15 for males).

## 4. Discussion

This study showed that patients with rheumatic disease had a two-fold increase in migraine prevalence (34%) and a three-fold increase in neuropathic pain prevalence (21.5%) compared to the general population. The prevalence of migraines was highest in patients with PsA and the prevalence of neuropathic pain was highest in patients with SpA. The migraine prevalence was also higher in patients treated with TNF-alpha inhibitors and in patients with RA accompanied by high inflammation.

Previous studies have suggested that migraines were linked to CIRDs [22,23]. We hypothesized that such a link might be explained by systemic inflammation, which might potentiate the neurogenic inflammation associated with migraines, as suggested in multiple sclerosis [10,24,25]. In the present study, systemic inflammation was never associated with migraine. This could be explained by the low levels of CRP observed (median 3.0 [IQR: 1.3–6.0] mg/L), due to the overall low disease activity.

We found that, among patients with RA, the use of TNF-alpha inhibitors was associated with an elevated risk of migraine. This result was quite surprising, because TNF-alpha has been shown to participate in the physiopathology of migraine, and thus, blocking it should, theoretically, reduce the frequency of migraines. There are two potential explanations for our findings. First, patients who used TNF-alpha inhibitors had highly active disease, and our results tended to show that high disease activity was associated with migraine occurrence. Alternatively, this effect might be a paradoxical effect; for example, TNF-alpha inhibitors aggravated psoriasis, when they should have improved it. Moreover, Soubrier et al. previously described neurological symptoms in patients treated with TNF-alpha inhibitors, particularly etanercept [26]. More recently, Kumar et al. recalled that TNF-alpha inhibitors could induce central inflammation and cause central nervous system demyelination. These contradictory effects are mediated by two different forms of TNF-alpha: A soluble form, which acts mainly on the TNF type-1 receptor (TNFR1), and a transmembranous form, which acts on the TNF type-2 receptor (TNFR2). TNFR1 binding leads to apoptosis and chronic inflammation; conversely, TNFR2 binding promotes cell survival, resolves inflammation, and induces remyelination. TNFR2 is abundant in the CNS; therefore, TNF-alpha inhibitors might promote demyelination by blocking the anti-inflammatory and regenerative effects of the transmembranous form of TNF-alpha on TNFR2. In fact, Kumar et al. suggested that there might be a class effect with TNF-alpha inhibitors, based on an analysis of 56 patients [27].

We found that, in our cohort, sulfasalazine treatment was clearly associated with a two-fold higher risk of neuropathic pain compared to the general population. Some cases have been described previously, where patients treated with sulfasalazine experienced burning in the extremities, particularly the soles of the feet, related to neurotoxicity, peripheral neuropathy, or axonal neuropathy [28,29,30]. The effects of sulfasalazine on neuropathic pain have not been sufficiently studied. A previous study in animals suggested that sulfasalazine treatment might be useful for treating nociceptive alterations in patients with diabetes [31]. Another study showed that sulfasalazine had the potential to act as a neuroprotective agent through its action on N-methyl-D-aspartate (NMDA) receptors [32].

It is important to note that neuropathic pain has been shown to be associated with a more severe disease course in RA. Indeed, a recent study showed that the probability of remission was very low in patients presenting neuropathic pain [33]. The same trend is present in the study, with a significantly more active disease among patients with neuropathic pain (Table 6). In addition, neuropathic pain and migraine were associated. As risk factors (age, sex, anxiety, anti-TNF treatments) do not similarly affect migraine and neuropathic pain, this suggests that these two pain syndromes do not rely on the same mechanisms. As expected, we found that migraines were correlated with a young age and high anxiety. Anxiety also occurs frequently in patients with fibromyalgia. Nearly 6% of our patients had a previous diagnosis of fibromyalgia, particularly those with spondyloarthritis. However, we found no significant association between fibromyalgia and migraine or neuropathic pain. These results rather contradicted findings in the literature [34,35,36]. Akdag Uzun et al. concluded that fibromyalgia occurred with greater frequency in patients with migraine, which supported a hypothesis that dopamine might play a role [37]. However, our study was not designed to analyze the association between fibromyalgia and migraine. In our study, fibromyalgia was not clinically researched, and the diagnosis was solely based on self-reports from patients. Therefore, we could not rule out the possibility that we might have underestimated the percentage of patients with fibromyalgia, as the prevalence was higher in several studies [38,39]. In addition, patients with migraine commonly experience pain catastrophizing, which is related to migraine severity and to anxiety [40]. In the present study, pain catastrophizing was not significantly associated with migraine or neuropathic pain. Regardless of the rheumatic disease, the catastrophizing score was always higher in patients with migraine or neuropathic pain, compared to the general population, but the difference never reached significance.

The main limitation of this study was that the patient sample was not highly representative of the population of patients with rheumatic disease. In most participants, the disease was well controlled, with a low disease activity and a low CRP level. This potential bias might be explained by the time required to complete the entire survey, on average between 15 and 30 min, including the physician-administered and self-administered questionnaires. This consultation time for completing the survey might only have been carried out with patients that were essentially well. In patients with high disease activity, the consultation time was mainly used to achieve remission by changing treatments, administering articular corticosteroid injections, and taking blood samples, rather than examining patients for the presence of migraine, neuropathic pain, or other comorbidities. However, currently, remission is the rule in rheumatic disease management; thus, the majority of patients had well-controlled rheumatic disease. Another limit was the absence of a contemporary control group. Although we had the opportunity to compare the prevalence of migraine and neuropathic pain to the results from studies conducted in large and representative samples of the French general population, these studies were conducted more than 10 years earlier.

In conclusion, this study found a high prevalence of migraine and neuropathic pain in a sample of patients with rheumatic disease. This finding should raise the awareness of the value of examining patients with CIRDs accompanied by residual pain for the presence of migraine or neuropathic pain, despite adequate clinical control of the rheumatic disease.

## Figures and Tables

**Figure 1 jcm-09-01890-f001:**
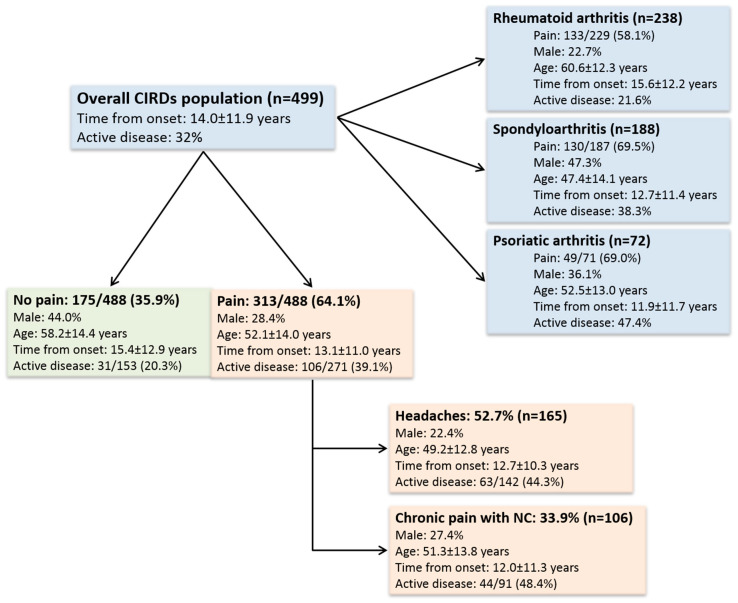
Distribution of rheumatic diseases, migraine, and neuropathic pain among study participants. CIRDs: Chronic inflammatory rheumatic diseases; NC: Neuropathic characteristics. The number in the denominator corresponds to the number of patients with available data.

**Table 1 jcm-09-01890-t001:** Characteristics of patients with chronic inflammatory rheumatic diseases.

Characteristic	All Patients N = 499	RA N = 238	SpA N = 188	PsA N = 72	*p* Value
Age, years [range]	54.5 ± 14.4 [18–87]	60.6 ± 12.3 [27–87]	47.4 ± 14.1 [18–85]	52.5 ± 13.0 [22–87]	<0.001
Men, *n* (%)	170 (34.1%)	54 (22.7%)	89 (47.3%)	26 (36.1%)	<0.001
Smokers, *n* (%)	104 (23.2%)	42 (17.6%)	45 (23.9%)	17 (23.6%)	0.49
Fibromyalgia, *n* (%)	32 (6.4%)	9 (3.7%)	16 (8.5%)	7 (9.7%)	0.14
Obesity, *n* (%)	95 (19.0%)	39 (16.4%)	33 (17.6%)	22 (30.6%)	0.008
Osteoarthritis, *n* (%)	89	47	26	16	0.28
Gougerot-Sjogren, *n* (%)	10 (2.0%)	9 (3.7%)	0 (0.0%)	1 (1.4%)	0.049
Depression, *n* (%)	41 (8.2%)	17 (7.1%)	20 (10.6%)	4 (5.6%)	0.46
Disease duration, years	11.2 [4.9–20.0]	14.0 [6.1–21.1]	9.0 [4.5–18.2]	9.5 [4.0–15.3]	0.01
Active disease, *n* (%)		42/194 (21.6%)	70/183 (38.3%)	27/57 (47.4%)	
DAS28 ESR score		2.34 ± 1.23		2.39 ± 1.38	
BASDAI, mm			33.6 ± 20.2	33.8 ± 20.5	
CRP, mg/L	3.0 [1.3–6.0]	3.0 [1.2–6.5]	2.9 [1.2–4.7]	3.0 [2.4–7.7]	0.11
HAQ score	0.58 ± 0.60	0.64 ± 0.67	0.52 ± 0.53	0.52 ± 0.53	0.09
HAD Anxiety score	7.8 ± 4.0	7.7 ± 4.2	8.1 ± 3.7	7.8 ± 4.0	0.81
PCS score	15.5 ± 11.5	14.1 ± 11.5	16.8 ± 11.3	15.8 ± 11.3	0.22
HAD Depression score	5.3 ± 3.9	5.0 ± 3.6	5.7 ± 4.3	4.9 ± 3.5	0.94
Treatments					
Pain treatment, *n* (%)	112 (22.4%)	50 (21.0%)	38 (20.2%)	24 (33.3%)	0.11
NSAID, *n* (%)	108 (21.6%)	30 (12.6%)	61 (32.4%)	17 (23.6%)	<0.001
Corticosteroid, *n* (%)	53 (10.6%)	46 (19.3%)	5 (2.7%)	2 (2.8%)	<0.001
Methotrexate, *n* (%)	234 (46.9%)	163 (68.4%)	34 (18.1%)	36 (50.0%)	<0.001
Sulfasalazine, *n* (%)	23 (4.6%)	7 (2.9%)	10 (5.3%)	6 (8.3%)	0.26
Leflunomide, *n* (%)	19 (3.8%)	15 (6.3%)	3 (1.6%)	1 (1.4%)	0.051
Anti-TNF-alpha, *n* (%)	241 (48.3%)	77 (32.4%)	130 (69.1%)	33 (45.8%)	<0.001
Hydroxychloroquine, *n* (%)	59 (11.8%)	57 (23.9%)	1 (0.05%)	0 (0.0%)	
Tocilizumab, *n* (%)	23 (4.6%)	23 (9.7%)	0 (0.0%)	0 (0.0%)	
Abatacept, *n* (%)	7 (1.4%)	7 (2.9%)	0 (0.0%)	0 (0.0%)	
Rituximab, *n* (%)	288 (5.6%)	27 (11.3%)	0 (0.0%)	0 (0.0%)	
Ustekinumab, *n* (%)	7/499 (1.4%)	0/238 (0.0%)	5 (2.7%)	2 (2.8%)	
Strict Migraine, *n* (%)	64/484 (13.2%)	23/203 (11.3%)	26/186 (14.0%)	15/71 (21.1%)	0.12
Strict + probable migraine, *n* (%)	165/484 (34.0%)	62/226 (27.4%)	71/186 (38.2%)	32/71 (45.1%)	0.02
Neuropathic pain, *n* (%)	106/493 (21.5%)	41/233 (17.6%)	50/187 (26.7%)	14/72 (19.4%)	0.02

RA: Rheumatoid arthritis; SpA: Spondyloarthritis; PsA: Psoriatic arthritis; DAS28: Disease activity in 28 joints, ESR: Erythrocyte sedimentation rate; CRP: C-reactive protein; HAQ: Health assessment questionnaire; BASDAI: Bath ankylosing spondylitis disease assessment index; HAD: Hospital anxiety and depression; PCS: Pain catastrophizing scale; NSAID: Non-steroid anti-inflammatory drug. Quantitative values are expressed as the mean ± standard deviation or the median [interquartile range], unless otherwise indicated.

**Table 2 jcm-09-01890-t002:** Parameters associated with occurrence of migraine, strict and probable, in inflammatory rheumatic diseases.

Parameter	Patients with Migraine *n* = 165	Patients without Migraine *n* = 324	Univariate *p*-Value OR [95%CI]	Multivariable *p*-Value OR [95%CI]
Age, years [range]	49.2 ± 12.8 [20–82]	56.7 ± 14.7 [18–87]	*p* < 0.001 OR = 0.96 [0.95–0.97]	*p* = 0.002 OR = 0.97 [0.95–0.99]
Female, *n* (%)	128/165 (77.6%)	190/319 (59.6%)	*p* < 0.001 OR = 2.34 [1.54–3.64]	*p* < 0.001 OR = 3.65 [2.01–6.84]
Smokers, *n* (%)	39 (26.7%)	64 (22.2%)	*p* = 0.78 OR = 1.03 [0.81–1.31]	
VAS for disease activity, mm	34.0 ± 25.7	25.4 ± 24.0	*p* < 0.001 OR = 1.01 [1.00–1.02]	*p* = 0.44 OR = 1.00 [0.99–1.02]
CRP, mg/L	3.0 [1.4–6.3]	2.9 [1.2–6.0]	*p* = 0.90 OR = 1.0 [0.98–1.02]	
Disease duration, years	12.7 [4.8–18.8]	12.6 [5.0–20.1]	*p* = 0.08 OR = 0.99 [0.96–1.00]	*p* = 0.85 OR = 1.00 [0.98–1.03]
HAD Anxiety score	9.5 ± 4.0	6.85 ± 3.6	*p* < 0.001 OR = 1.20 [1.14–1.27]	*p* = 0.003 OR = 1.13 [1.04–1.23]
HAD Depression score	6.4 ± 4.2	4.66 ± 3.6	*p* < 0.001 OR = 1.12 [1.07–1.18]	*p* = 0.46 OR = 1.03 [0.95–1.12]
PCS score	17.8 ± 11.4	13.9 ± 11.2	*p* = 0.001 OR = 1.03 [1.01–1.05]	*p* = 0.95 OR = 1.00 [0.97–1.02]
Pain treatment *, *n* (%)	46/165 (27.9%)	60/319 (19%)	*p* = 0.02 OR = 1.67 [1.07–2.59]	*p* = 0.20 OR = 1.56 [0.79–3.06]
Anti-TNF-alpha treatment *, *n* (%)	96/156 (58.2%)	143/319 (44.8%)	*p* = 0.005 OR = 1.71 [1.17–2.51]	*p* = 0.017 OR = 1.90 [1.13–3.25]
Neuropathic pain, *n* (%)	62/165 (37.6%)	43/318 (13.5%)	*p* < 0.001 OR = 3.85 [2.46–6.06]	*p* = 0.005 OR = 2.49 [1.32–4.71]

***** Other treatments did not show a significant difference between groups. VAS: Visual analogue scale; CRP: C-reactive protein; HAD: Hospital anxiety and depression; PCS: Pain catastrophizing scale. Quantitative values are expressed as the mean ± standard deviation or the median [interquartile range], unless otherwise indicated.

**Table 3 jcm-09-01890-t003:** Parameters associated with the occurrence of migraine, strict and probable, in rheumatoid arthritis.

Parameters	Patients with Migraine *n* = 62	Patients without Migraine *n* = 164	Univariate *p* Value OR [95%CI]	Multivariable *p* Value OR [95%CI]
Age, years [range]	56.2 ± 11.0 [30–76]	61.8 ± 12.5 [27–87]	*p* = 0.003 OR = 0.96 [0.94–0.99]	*p* = 0.03 OR = 0.96 [0.93–0.99]
Female	54/62 (87.1%)	121/164 (73.8%)	*p* = 0.001 OR = 2.4 [1.1–5.8]	*p* = 0.59 OR = 1.42 [0.43–5.67]
Smokers	13/56 (23.2%)	28/150 (18.7%)	*p* = 0.15 OR = 1.30 [0.91–1.84]	
DAS28 ESR score	2.74 ± 1.19	2.24 ± 1.33	*p* = 0.02 OR = 1.37 [1.05–1.80]	*p* = 0.08 OR = 1.35 [0.97–1.89]
CRP, mg/L	3.0 [1.8–6.3]	2.9 [1.0–6.6]	*p* = 0.42 OR = 1.02 [0.97–1.06]	
Disease duration, years	12.4 [5.0–20.3]	14.7 [7.0–21.7]	*p* = 0.26 OR = 0.99 [0.96–1.01]	
Erosive RA	26/62 (41.9%)	78/164 (47.5%)	*p* = 0.45 OR = 0.80 [0.44–1.43]	
RF and/or ACPA positive	49/62 (79.0%)	127/160 (77.4%)	*p* = 0.80 OR = 1.01 [0.55–2.30]	
HAD Anxiety score	9.3 ± 4.6	6.9 ± 3.66	*p* < 0.001 OR = 1.16 [1.08–1.25]	*p* = 0.83 OR = 0.99 [0.86–1.12]
HAD Depression score	6.5 ± 4.2	4.4 ±3.27	*p* < 0.001 OR = 1.16 [1.07–1.26]	*p* = 0.03 OR = 1.14 [1.01–1.30]
PCS score	17.2 ± 12.0	12.5 ±10.9	*p* = 0.02 OR = 1.04 [1.01–1.07]	*p* = 0.48 OR = 1.01 [0.98–1.05]

All treatments showed no significant difference between groups. DAS28: Disease activity in 28 joints, ESR: Erythrocyte sedimentation rate; CRP: C-reactive protein; RA: Rheumatoid arthritis; RF: Rheumatoid factor; ACPA: Anti-citrullinated peptides’ antibodies; HAD: Hospital anxiety and depression; PCS: Pain catastrophizing scale. Quantitative values are expressed as the mean ± standard deviation, the median [interquartile range], or the number (%), as indicated.

**Table 4 jcm-09-01890-t004:** Parameters associated with the occurrence of migraine, strict and probable, in spondyloarthritis.

Parameter	Patients with Migraine *n* = 71	Patients without Migraine *n* = 115	Univariate *p* Value OR [95%CI]	Multivariable *p* Value OR [95%CI]
Age, years [range]	44.8 ± 12.9 [20–82]	48.8 ± 14.6 [18–85]	*p* = 0.06 OR= 0.98 [0.96–1.00]	*p* = 0.15 OR= 0.98 [0.96–1.01]
Female	49/71(69.0%)	49/115 (57.4%)	*p* < 0.001 OR= 3.0 [1.62–5.70]	*p* = 0.01 OR = 2.5 [1.3–5.2]
Smokers	22/62 (35.5%)	26/101 (25.7%)	*p* = 0.55 OR= 1.12 [0.76–1.66]	
BASDAI, mm	41.5 ± 19.4	28.8 ± 19.4	*p* < 0.001 OR = 1.03 [1.02–1.05]	*p* = 0.004 OR= 1.03 [1.01–1.05]
CRP, mg/L	2.9 [1.0–4.3]	2.9 [1.4–4.7]	*p* = 0.66 OR= 0.99 [0.96–1.02]	
Disease duration, years	6.6 [4.7–15.4]	9.9 [3.7–18.5]	*p* = 0.13 OR= 0.98 [0.95–1.00]	
HAD Anxiety score	9.5 ± 3.6	7.1 ± 3.5	*p* < 0.001 OR = 1.20 [1.10–1.31]	*p* = 0.004 OR = 1.19 [1.06–1.34]
HAD Depression score	6.4 ± 4.4	5.2 ± 4.2	*p* = 0.06 OR= 1.07 [0.99–1.14]	*p* = 0.23 OR= 0.94 [0.84–1.04]
PCS score	18.4 ± 10.6	15.7 ± 11.8	*p* = 0.15 OR= 1.02 [0.99–1.05]	

All treatments showed no significant difference between groups. BASDAI: Bath ankylosing spondylitis disease assessment index; CRP: C-reactive protein; HAD: Hospital anxiety and depression; PCS: Pain catastrophizing scale. Quantitative values are expressed as mean ± standard deviation, the median [interquartile range], or the number (%), as indicated.

**Table 5 jcm-09-01890-t005:** Comparison of the 165 patients with migraine.

Parameter	All Patients N = 165	RA N = 62	SpA N = 71	PsA N = 32	*p* Value
Age, years [range]	49.2 ± 12.8 [20–82]	56.2 ± 11.0 [30–76]	44.8 ± 12.9 [20–82]	45.5 ± 9.8 [22–63]	*p* < 0.001
Men, *n* (%)	37/165 (22.4%)	8/62 (12.9%)	22/71 (31.0%)	7/32 (21.9%)	*p* = 0.04
HIT-6 > 55, *n* (%)	120/165 (72.7%)	46/62 (74.2%)	49/71 (69.0%)	25/32 (78.1%)	*p* = 0.60
Headaches ≥ 15 d/month, *n* (%)	18/149 (12.1%)	9/53 (17.0%)	7/69 (10.1%)	2/27 (7.4%)	*p* = 0.28
Headaches 8–14 d/month, *n* (%)	22/149 (14.8%)	4/53 (7.5%)	12/69 (17.4%)	6/27 (22.2%)	
Headaches < 8 d/month, *n* (%)	109/149 (73.2%)	40/53 (75.5%)	50/69 (72.5%)	19/27 (70.4%)	

RA: Rheumatoid arthritis; SpA: Spondyloarthritis; PsA: Psoriatic arthritis; HIT-6: Six-item Headache Impact Test. Quantitative values are expressed as the mean ± standard deviation or the median [interquartile range], unless otherwise indicated.

**Table 6 jcm-09-01890-t006:** Parameters associated with the occurrence of neuropathic pain in rheumatic disease.

Parameter	Patients with Neuropathic Pain *n* = 106	Patients without Neuropathic Pain *n* = 392	Univariate *p* Value OR [95%CI]	Multivariable *p* Value OR [95%CI]
Age, years [range]	51.3 ± 13.8 [22–82]	55.4 ± 14.5 [18–87]	*p* = 0.01 OR = 0.96 [0.95–0.98]	*p* = 0.26 OR = 1.00 [0.96–1.01]
Female	77/106 (72.6%)	252/392 (64.3%)	*p* = 0.11 OR = 1.48 [0.93–2.40]	
Smokers	29/99 (29.3%)	74/349 (21.2%)	*p* = 0.87 OR = 1.02 [0.78–1.33]	
VAS disease activity, mm	39.4 ± 26.0	24.9 ± 23.6	*p* < 0.001 OR = 1.02 [1.01–1.03]	*p* = 0.02 OR = 1.00 [1.00–1.03]
CRP, mg/L	3.0 [1.3–5.7]	3.0 [1.3–6.0]	*p* = 0.91 OR = 1.00 [0.97–1.02]	
Disease duration, years	7.9 [4.5–17.9]	12.6 [5.1–20.3]	*p* = 0.06 OR = 0.98 [0.96–1.00]	*p* = 0.06 OR = 1.01 [0.94–1.03]
HAQ	0.86 ± 0.64	0.51 ± 0.57	*p* < 0.001 OR = 2.44 [1.73–3.47]	*p* = 0.01 OR = 1.43 [1.18–3.50]
HAD Anxiety score	9.4 ± 3.8	7.4 ± 3.9	*p* < 0.001 OR = 1.13 [1.07–1.19]	*p* = 0.13 OR = 1.00 [0.98–1.17]
HAD Depression score	6.9 ± 4.1	4.8 ± 3.7	*p* < 0.001 OR = 1.14 [1.08–1.20]	*p* = 0.47 OR = 1.13 [0.94–1.13]
PCS score	18.6 ± 10.9	14.6 ± 11.5	*p* = 0.003 OR= 1.03 [1.01–1.05]	*p* = 0.71 OR = 1.00 [0.97–1.02]
NSAID Treatment *	30/106 (28.3%)	78/392 (20.0%)	*p* = 0.06 OR = 1.59 [0.96-2.58]	*p* = 0.13 OR = 0.97 [0.85-3.21]
Pain Treatment *	35/106 (33.0%)	77/392 (19.6%)	*p* = 0.004 OR = 2.01 [1.25-3.23]	*p* = 0.19 OR = 1.56 [0.79-3.09]
Sulfasalazine Treatment *	9/106 (8.5%)	14/392 (3.6%)	*p* = 0.04 OR = 2.51 [1.02–5.89]	*p* = 0.04 OR = 1.90 [1.05–12.7]

* Other treatments did not show a significant difference between groups. VAS: Visual analogue scale; CRP: C-reactive protein; HAD: Hospital anxiety and depression; HAQ: Health Assessment Questionnaire; PCS: Pain catastrophizing scale; NSAID: Non-steroid anti-inflammatory drug. Quantitative values are expressed as mean ± standard deviation, the median [interquartile range], or the number (%), as indicated.

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
