# Peer review of "Prevalence of Migraine and Neuropathic Pain in Rheumatic Diseases"

_jcm, 2020, doi:10.3390/jcm9061890_

Round 1

Reviewer 1 Report

I read with interest this work which shows the highest prevalence of migraine and neuropathic pain in patients with chronic inflammatory joint disease. The topic is certainly interesting and topical. However, I believe that the work needs to be improved before it can be accepted for publication.

Abstract

It is stated "High disease activity was also a risk factor in AS". It is assumed that high disease activity was also a risk factor for migraine in another condition. Which condition?

What does "Systemic inflammation" stand for? High CRP or ESR? Please clarify as it is then done in the text.

Introduction

The definition "chronic inflammatory rheumatism" should be changed to "chronic inflammatory rheumatic diseases" throughout the paper.

The authors should better explain the rationale of including neuropathic pain, neuropathic pain and migraine in the same study, somehow considering some affinity. In accordance with the recent reclassification of pain by IASP apparently have nothing to do with each other and are therefore separate entities.

So why should they be considered in any way related? Review the introduction in this sense.

Linked to what has just been said, I would speak of neuropathic pain features in the text because, again in accordance with the IASP classification of pain, to define neuropathic pain it is necessary to demonstrate the nerve injury with instrumental techniques.

Methods

Page 2, lines 64-67. It is said that patients with RA, PsA and AS have been enrolled. Then the ASAS criteria are also mentioned for non-radiographic axial forms and for peripheral forms. Were patients with a generic axial SpA or only those with AS included? Please clarify.

Results

Why are there three Tables 3? Please adjust.

Page 4 line 138. The DAS28 is expressed in mg/l! Maybe you should replace mg/l with standard deviation.

The proportion of patients with FMS is surprisingly low (6%), especially in PsA (9.7%). From previous data the prevalence of FMS in PsA is around 18% (see the works of Di Carlo et al J Rheumatol 2017,44(3):279-85, Brikman et al J Rheumatol 2016,43(99):1749-54). In the Discussion the authors then explain that it was a self-reported, and perhaps underestimated diagnosis.

Discussion

Discuss the fact that in the RA the presence of neuropathic pain is also present in the early stages of the disease and then influences the achievement of remission (Salaffi F, Di Carlo M, Carotti M, Sarzi-Puttini P. The Effect of Neuropathic Pain Symptoms on Remission in Patients with Early Rheumatoid Arthritis. Curr Rheumatol Rev. 2019;15(2):154-161. doi: 10.2174/1573397114666180806142814).

Page 9, line 256. Among the papers documenting the association between neuropathic pain and FMS in PsA, there is a recent work by Di Carlo and colleagues (which should be mentioned, Di Carlo M, Muto P, Benfaremo D, Luchetti MM, Atzeni F, Salaffi F. The neuropathic pain features in Psoriatic Arthritis: a cross-sectional evaluation of prevalence and associated factors. J Rheumatol 2019; doi: 10.3899/jrheum.190906) which shows how FMS is the major predictor of a high PainDetect score. Excluding patients with FMS, HAQ is the main predictor of neuropathic pain. The multivariate analysis should be repeated to include HAQ since it has been collected.

Figure 1 to improve in quality, it is difficult to read.

Author Response

Letter to the Editor of Journal of Clinical Medicine, Ms.Siiri Ma

                                                             Clermont-Ferrand, June, 5th 2020

            Dear Madam,

Please find attached the revised version of an article we are proposing for publication, entitled “Prevalence of migraine and neuropathic pain in rheumatic diseases.”, authors S Mathieu et al, Manuscript ID: jcm-812115.

We wish to thank you and the reviewers for your interest in our manuscript. We have taken into account the reviewers’ comments in the revised version (please see answers to reviewers below and revised version attached).

We hereby confirm that the article has not been published and is not under consideration for publication elsewhere. There were no financial support or other benefits from commercial sources for the work reported on in the manuscript, or any other financial interests that any of the authors may have, which could create a potential conflict of interest or the appearance of a conflict of interest with regard to the work.

The manuscript has been read and approved by all authors.

Best regards,

Sylvain Mathieu, corresponding author

Dr. Sylvain Mathieu,

Service de rhumatologie, CHU Gabriel Montpied

58 rue Montalembert, 63000 Clermont-Ferrand, France.

Telephone (33) 4 73751488 Fax (33) 4 73751489

Email [email protected]

Reviewer 1. Comments and Suggestions for Authors

I read with interest this work which shows the highest prevalence of migraine and neuropathic pain in patients with chronic inflammatory joint disease. The topic is certainly interesting and topical. However, I believe that the work needs to be improved before it can be accepted for publication.

Abstract

It is stated "High disease activity was also a risk factor in AS". It is assumed that high disease activity was also a risk factor for migraine in another condition. Which condition?

Response: We completely agree that this sentence is not very clear. We wanted to write that in addition to “high level of anxiety, female sex, young age, and TNF‑alpha inhibitor treatment”, noted in the preceding sentence as risk factors for migraine, high disease activity was also a risk factor for migraine but only in the sub-group of patients with AS.

To improve clarity, we have now written: “Besides, high disease activity was a risk factor in AS”.

What does "Systemic inflammation" stand for? High CRP or ESR? Please clarify as it is then done in the text.

Response: Systemic inflammation corresponded to a high level of C reactive protein (CRP). Systemic inflammation was defined in the main text but not in the abstract. We have now written: “Blood CRP level was not significantly associated with migraine.”

Introduction

The definition "chronic inflammatory rheumatism" should be changed to "chronic inflammatory rheumatic diseases" throughout the paper.

Response: We agree with this suggestion and have change "chronic inflammatory rheumatism" to "chronic inflammatory rheumatic diseases (CIRDs)" throughout the paper.

The authors should better explain the rationale of including neuropathic pain, neuropathic pain and migraine in the same study, somehow considering some affinity. In accordance with the recent reclassification of pain by IASP apparently have nothing to do with each other and are therefore separate entities.

So why should they be considered in any way related? Review the introduction in this sense.

Response: To address this point, we have added two sentences in the introduction: “Although there is no pathophysiological relationship between migraine and neuropathic pain, it has been shown in a previous study focusing on pain in patients with multiple sclerosis that these two pain entities were associated [10]. As there was a rationale for an increased prevalence of both migraine and neuropathic pain in CIRDs, we decided to study both in the same study.”

Linked to what has just been said, I would speak of neuropathic pain features in the text because, again in accordance with the IASP classification of pain, to define neuropathic pain it is necessary to demonstrate the nerve injury with instrumental techniques.

Response: Neuropathic pain definition has been added in the introduction ("pain arising as a direct consequence of a lesion or disease affecting the somatosensory system) and clarifications about the definition used have been added in the methods section. It is now written: “A grading system is recommended for neuropathic pain’s diagnosis [19]. Although this is the gold standard, such grading system can be difficult to use for non-specialists and using validated screening questionnaires in large epidemiological studies is commonly accepted. In the present study, the subjects answered the 2 questions (including seven items) from the DN4 interview questionnaire regarding the characteristics of their pain [13]. A score of 1 was given to each positive item and a score of 0 to each negative item. The total score was calculated as the sum of the 7 items. Respondents with a total score ≥3 were considered to have neuropathic pain characteristics [13]. Initially, the DN4 interview questionnaire was validated as a clinician administered questionnaire. A complementary validation was made, and the results of the self-reported and clinician-administered questions for each of the 7 items showed excellent consistency (κ coefficients with a 95% confidence interval [CI] of 0.82–0.95; P < .001). The self-administered DN4 questionnaire had a sensitivity of 81.6% and a specificity of 85.7% for an optimal cut-off score of 3 out of 7, being similar to those observed in the initial study using a clinician-administered version of the questionnaire [20].”

Methods

Page 2, lines 64-67. It is said that patients with RA, PsA and AS have been enrolled. Then the ASAS criteria are also mentioned for non-radiographic axial forms and for peripheral forms. Were patients with a generic axial SpA or only those with AS included? Please clarify.

Response: We decided to include all patients that visited our Rheumatology department for spondyloarthropathy, which corresponded to radiographic ankylosing spondylitis that fulfilled the New York criteria, to non-radiographic axial spondyloarthropathy that fulfilled ASAS axial criteria and to peripheral spondyloarthropathye that fulfilled ASAS peripheral criteria. A new term “spondyloarthritis (SpA)” has been used in the revised manuscrit to define all these patients.

Results

Why are there three Tables 3? Please adjust.

Response: The tables are now numbered from 1 to 6 (a new table has been added concerning migraine, after the suggestion of reviewer #2).

Page 4 line 138. The DAS28 is expressed in mg/l! Maybe you should replace mg/l with standard deviation.

Response: Thank you very much for correcting this mistake!

The proportion of patients with FMS is surprisingly low (6%), especially in PsA (9.7%). From previous data the prevalence of FMS in PsA is around 18% (see the works of Di Carlo et al J Rheumatol 2017,44(3):279-85, Brikman et al J Rheumatol 2016,43(99):1749-54). In the Discussion the authors then explain that it was a self-reported, and perhaps underestimated diagnosis.

Response: We completely agree that definition of fibromyalgia in our study was based on patient interrogation and therefore was a self-reported information. We hypothetized that this choice of data collection might have underestimated the proportion of patients with fibromyalgia and could explain our lower percentage compared with the medical literature. The low prevalence of reported fibromyalgia was already discussed. We have added the suggested references to highlight this point. 

Discussion

Discuss the fact that in the RA the presence of neuropathic pain is also present in the early stages of the disease and then influences the achievement of remission (Salaffi F, Di Carlo M, Carotti M, Sarzi-Puttini P. The Effect of Neuropathic Pain Symptoms on Remission in Patients with Early Rheumatoid Arthritis. Curr Rheumatol Rev. 2019;15(2):154-161. doi: 10.2174/1573397114666180806142814).

Response: Indeed, neuropathic pain can be present early in the disease course and is associated with a lower rate of remission. We have now quoted the suggested reference. 

Page 9, line 256. Among the papers documenting the association between neuropathic pain and FMS in PsA, there is a recent work by Di Carlo and colleagues (which should be mentioned, Di Carlo M, Muto P, Benfaremo D, Luchetti MM, Atzeni F, Salaffi F. The neuropathic pain features in Psoriatic Arthritis: a cross-sectional evaluation of prevalence and associated factors. J Rheumatol 2019; doi: 10.3899/jrheum.190906) which shows how FMS is the major predictor of a high PainDetect score. Excluding patients with FMS, HAQ is the main predictor of neuropathic pain. The multivariate analysis should be repeated to include HAQ since it has been collected.

Response: Thank you for this suggestion. We added this recent reference in the revised manuscript that reported an association between neuropathic pain and fibromyalgia in patients with psoriactic arthritis. HAQ has been added in Table 6 and the multivariate analysis has been repeated.

Figure 1 to improve in quality, it is difficult to read.

Response: We have tried to improve quality of figure 1 to make easier to read.

Reviewer 2 Report

Review of Manuscript; Prevalence of migraine and neuropathic pain in rheumatic diseases

This is a study with a large number of subjects and which addresses an important aspect of CIR. There are a number of aspects of the report that do need consideration and will be outlined below.

1. The manuscript contains a large number of abbreviations. This produces significant difficulty for the reader and makes the manuscript somewhat arduous to read. Suggest adding a glossary of abbreviations that the reader can access readily and easily to expedite the reader’s progress.

2. The report is based on use of a number of questionnaires (QS) which include 1. rheumatic disease factors, 2. headache QS, 3. neuropathic pain QS, 4. HAQ, 5. HAD, 6. BASDAI, 7. PCS. In the Methods section the authors state that this QS could be completed in 15 minutes. While some subjects could complete the QS in 15 minutes, I doubt that everyone could. I suspect there is a range of times needed for completion that could be much longer. When filling out QS, subjects may become bored, fatigued or uninterested and simply stop providing the intensity of concentration necessary for accurate answers. Was the study QS administered to controls with similar backgrounds to the test subjects to determine how much time was required to complete the QS? Were the QS checked bystudy investigators for accuracy?

3. How much of the data employed was strictly from the QS and how much was from the investigators direct contact with the subject as a result of medical visit?

4. With regard to the problem of headache, the prevalence of migraine for strict migraine (SM) was about the same as the historical controls (Lanteri-2005) for RA and the major difference was for the smaller PsA group. The AS group was intermediate. The probable migraine (PM) made the largest contribution to the differences reported for the entire CIR population. According to Lanteri et al (2005) the strict migraine group has almost twice the prevalence of severe headaches which suggests a greater headache “burden” for the SM group. Is combination of SM and PM groups justified?

5. There is no information given about the headache burden or impact. Headache is measured in terms of frequency, duration and intensity. Is There a difference in the headache burden among the three CIR groups? Con sider comparing the prevalence of Chronic Migraine, Frequent Migraine and Infrequent Migraine among the groups.

6. The occurrence of neuropathic pain is significant. I there a correlation between neuropathic pain and migraine?

6. There is no contemporary control group. The controls are related to studies that are 12-13 yeares old. The comparison to relatively old control material should be discussed. The study would have been significantly enhanced by acquiring data from a modest number of controls and comparing them to the older studies to confirm validity of data from the older studies.

7. The age range of the subjects is very broad. Considering two standard deviations from the mean of 54.5 years, the range is 26-82 years. Migraine prevalence is approximately 5% of prepubertal children and rises after puberty to approximately 30% of females and 13% of males between 30 and 50-55 years, then falling to 3-5% of the population after age 70. Thje prevalence of neuropathic pain, on the other hand, increases with age. The data might be better represented by presentation of quartiles.

8. Should a correction be applied for multiple tests?

Author Response

Letter to the Editor of Journal of Clinical Medicine, Ms.Siiri Ma

                                                                   Clermont-Ferrand, June, 5th 2020

            Dear Madam,

Please find attached the revised version of an article we are proposing for publication, entitled “Prevalence of migraine and neuropathic pain in rheumatic diseases.”, authors S Mathieu et al, Manuscript ID: jcm-812115.

We wish to thank you and the reviewers for your interest in our manuscript. We have taken into account the reviewers’ comments in the revised version (please see answers to reviewers below and revised version attached).

We hereby confirm that the article has not been published and is not under consideration for publication elsewhere. There were no financial support or other benefits from commercial sources for the work reported on in the manuscript, or any other financial interests that any of the authors may have, which could create a potential conflict of interest or the appearance of a conflict of interest with regard to the work.

The manuscript has been read and approved by all authors.

Best regards,

Sylvain Mathieu, corresponding author

Dr. Sylvain Mathieu,

Service de rhumatologie, CHU Gabriel Montpied

58 rue Montalembert, 63000 Clermont-Ferrand, France.

Telephone (33) 4 73751488 Fax (33) 4 73751489

Email [email protected]

Reviewer 2. Comments and Suggestions for Authors.

Review of Manuscript; Prevalence of migraine and neuropathic pain in rheumatic diseases

This is a study with a large number of subjects and which addresses an important aspect of CIR. There are a number of aspects of the report that do need consideration and will be outlined below.

  1. The manuscript contains a large number of abbreviations. This produces significant difficulty for the reader and makes the manuscript somewhat arduous to read. Suggest adding a glossary of abbreviations that the reader can access readily and easily to expedite the reader’s progress.

Response: We completely agree that addition of a glossary of abbreviations would improve the revised manuscript and make it easier to read. This has been done.

  1. The report is based on use of a number of questionnaires (QS) which include 1. rheumatic disease factors, 2. headache QS, 3. neuropathic pain QS, 4. HAQ, 5. HAD, 6. BASDAI, 7. PCS. In the Methods section the authors state that this QS could be completed in 15 minutes. While some subjects could complete the QS in 15 minutes, I doubt that everyone could. I suspect there is a range of times needed for completion that could be much longer. When filling out QS, subjects may become bored, fatigued or uninterested and simply stop providing the intensity of concentration necessary for accurate answers. Was the study QS administered to controls with similar backgrounds to the test subjects to determine how much time was required to complete the QS? Were the QS checked bystudy investigators for accuracy?

Response: Most of the subjects were able to complete the questionnaire in 15 minutes. Nonetheless, it was slightly longer for some of the participants. All the QS were checked for accuracy by one of the investigators after completion. These precisions have been added.

  1. How much of the data employed was strictly from the QS and how much was from the investigators direct contact with the subject as a result of medical visit?

Response: The pain questionnaires (BPI, headache, neuropathic pain and PCS), HAD and PCS were completed by the patients alone and checked by the investigator during the medical visit. Questionnaires concerning rheumatic disease factors were completed with the physician.

  1. With regard to the problem of headache, the prevalence of migraine for strict migraine (SM) was about the same as the historical controls (Lanteri-2005) for RA and the major difference was for the smaller PsA group. The AS group was intermediate. The probable migraine (PM) made the largest contribution to the differences reported for the entire CIR population. According to Lanteri et al (2005) the strict migraine group has almost twice the prevalence of severe headaches which suggests a greater headache “burden” for the SM group. Is combination of SM and PM groups justified?

Response: It is not possible to perform a direct comparison of the proportion of migraineurs in the historical cohort (lantéri 2005) and in the present study as age and sex are not similar. As migraine prevalence varies greatly with these two factors, we have conducted subgroup analysis to avoid such bias and we show that there still is a 2-fold migraine prevalence increase when taking age into account. As RA patients are older (60.6 ± 12.3), it tends to decrease the proportion of migraineurs in this subgroup (11.3% of strict migraine and 27.4% overall). For comparison, the prevalence was 5.3% and 11.1% overall in the study by Lantéri et al. in the subjects aged > 55 years. Thus, the two-fold increase is present in RA, PsA and AS when taking these confounding factors into account.

As noted by the reviewer, the intensity of migraine was higher in strict migraine compared to probable migraine in the study by Lantéri et al. Nonetheless, the conclusion of this study was “If probable and strict migraine are two phenotypic forms of the same entity, and the burdens inflicted on patients is comparable, similar treatments must be proposed for the two medical conditions. The same recommendations as for strict migraine may apply for probable migraine, including medical follow-up, prescription of specific treatments and drug intake as soon as possible after the beginning of an attack”. Pooling strict and probable migraine is commonly accepted in epidemiological studies. For example, according to the Global Burden of Disease Study 2016, the global age-standardised prevalence of migraine is 14.4% overall, with a 18% prevalence in France, which is closed to the 21% of strict + probable migraine (GBD2016 Lancet Neurol 2018). Thus, we have decided to do the same in our study.

  1. There is no information given about the headache burden or impact. Headache is measured in terms of frequency, duration and intensity. Is There a difference in the headache burden among the three CIR groups? Con sider comparing the prevalence of Chronic Migraine, Frequent Migraine and Infrequent Migraine among the groups.

Response: Considering headache impact, we had only written: “Among the 165 patients with migraine, 124 (75.2%) had HIT-6 scores above 55, which indicated that headache had an important impact on these patients.”

 We have now added a table to compare the headache burden among the three groups:

Table 5. Comparison of the 165 patients with migraine

Parameter

All patients

N = 165

RA

N = 62

AS

N = 71

PsA

N = 32

P value

Age, years

[range]

49.2 ± 12.8

[20-82]

56.2 ± 11.0

[30-76]

44.8 ± 12.9

[20-82]

45.5 ± 9.8

[22-63]

p<0.001

Men, n (%)

37/165 (22.4%)

8/62 (12.9%)

22/71 (31.0%)

7/32 (21.9%)

p=0.04

HIT-6 > 55, n (%)

120/165 (72.7%)

46/62 (74.2%)

49/71 (69.0%)

25/32 (78.1%)

p=0.60

Headaches ≥ 15 d/month, n (%)

18/149 (12.1%)

9/53 (17.0%)

7/69 (10.1%)

2/27 (7.4%)

p=0.28

Headaches 8-14 d/month, n (%)

22/149 (14.8%)

4/53 (7.5%)

12/69 (17.4%)

6/27 (22.2%)

Headaches < 8 d/month, n (%)

109/149 (73.2%)

40/53 (75.5%)

50/69 (72.5%)

19/27 (70.4%)

RA: rheumatoid arthritis; SpA: spondyloarthritis; PsA: psoriatic arthritis; HAD: HIT-6: six-item Headache Impact Test. Quantitative values are expressed as the mean ± standard deviation or the median [interquartile range], unless otherwise indicated.

  1. The occurrence of neuropathic pain is significant. I there a correlation between neuropathic pain and migraine?

Response: Neuropathic pain has been added in Table 2 to show the presence of correlation between neuropathic pain and migraine, as exposed page 10.  Two sentences have been added in the discussion concerning this point and that the mechanisms underlying these two symptoms are probably different.

  1. There is no contemporary control group. The controls are related to studies that are 12-13 yeares old. The comparison to relatively old control material should be discussed. The study would have been significantly enhanced by acquiring data from a modest number of controls and comparing them to the older studies to confirm validity of data from the older studies.

Response: We completely agree that ideally a control group should be contemporary to permit a relevant conclusion. We have now clearly noted this limit in the discussion, adding these sentences: “Another limit is the absence of a contemporary control group. Although we had the opportunity to compare the prevalence of migraine and neuropathic pain to the results from studies conducted in large and representative samples of the French general population, these studies were conducted more than 10 years earlier.”

  1. The age range of the subjects is very broad. Considering two standard deviations from the mean of 54.5 years, the range is 26-82 years. Migraine prevalence is approximately 5% of prepubertal children and rises after puberty to approximately 30% of females and 13% of males between 30 and 50-55 years, then falling to 3-5% of the population after age 70. Thje prevalence of neuropathic pain, on the other hand, increases with age. The data might be better represented by presentation of quartiles.

Response: The reviewer is right. We have taken into account the effect of age in the analysis of the data. Our data can be better represented with the range. Thus, as suggested, we have added the range in all tables.

  1. Should a correction be applied for multiple tests?

Response: We thank the reviewer for the helpful comment. As univariate analyses could be considered exploratory and principally helpful to determine covariates candidate to multivariable, we have chosen (i) to report all the individual p-values and confidence intervals, without doing any mathematical correction for distinct tests comparing two modalities (Rothman, K.J. (1990). No adjustments are needed for multiple comparisons. Epidemiology 1, 43–46. Feise, R.J. (2002) Do multiple outcome measures require p-value adjustment? BMC Med Res Methodol. 2:8) and (ii) paid on a specific attention to the magnitude of differences and to clinical relevance. This point is now stipulated in the methods. Furthermore, in tables 2, 3, 4 and 6, a majority of comparisons were significant with a P-value < 0.001, confirming that results should not be modified with a correction of the type I error.

Round 2

Reviewer 1 Report

The authors, since they have modified the multivariate analysis to include HAQ as suggested in Table 6, and since it is significant, should also report this finding in full in the results.
Then, to discuss this (HAQ and neuropathic pain features in PsA) in the light of recent literature data on the subject, they should take as a reference the suggested work (The neuropathic pain features in Psoriatic Arthritis: a cross-sectional evaluation of prevalence and associated factors. J Rheumatol 2019; doi: 10.3899/jrheum.190906).
Otherwise, the work seems to me to have improved overall.

Author Response

Letter to the Editor of Journal of Clinical Medicine, Ms.Siiri Ma

                                                            Clermont-Ferrand, June, 12th 2020

            Dear Madam,

Please find attached the new revised version of an article we are proposing for publication, entitled “Prevalence of migraine and neuropathic pain in rheumatic diseases.”, authors S Mathieu et al, Manuscript ID: jcm-812115.

We wish to thank you and the reviewers for your interest in our manuscript. We have taken into account the reviewers’ comments in the new revised version (please see answers to reviewers below and changes in colour in the new manuscript).

The manuscript has been read and approved by all authors.

Best regards,

Sylvain Mathieu, corresponding author

Dr. Sylvain Mathieu,

Service de rhumatologie, CHU Gabriel Montpied

58 rue Montalembert, 63000 Clermont-Ferrand, France.

Telephone (33) 4 73751488 Fax (33) 4 73751489

Email [email protected]

Reviewer 1. Comments and Suggestions for Authors

The authors, since they have modified the multivariate analysis to include HAQ as suggested in Table 6, and since it is significant, should also report this finding in full in the results.

Response: We have changed the results section and added the significant results of HAQ, as parameter associated with the occurrence of neuropathic pain in rheumatic diseases.

Then, to discuss this (HAQ and neuropathic pain features in PsA) in the light of recent literature data on the subject, they should take as a reference the suggested work (The neuropathic pain features in Psoriatic Arthritis: a cross-sectional evaluation of prevalence and associated factors. J Rheumatol 2019; doi: 10.3899/jrheum.190906).
Otherwise, the work seems to me to have improved overall.

Response: This reference has been added in the discussion to highlight the relation between HAQ and neuropathic pain.